# A comprehensive analysis of clinical, quality of life, and cost-effectiveness outcomes of key treatment options for benign prostatic hyperplasia

Bilal Chughtai[1]⊙, Sirikan Rojanasarot[2]⊙*, Kurt Neeser[3‡], Dmitry Gultyaev[3‡], Shuai Fu[3‡], Samir K. Bhattacharyya[2], Ahmad M. El-Arabi[4], Ben J. Cutone[2], Kevin T. McVary[4]⊙

1 Department of Urology, Weill Cornell Medicine, New York, New York, United States of America, 2 Boston Scientific, Marlborough, MA, United States of America, 3 Certara Evidence & Access, Lörrach, BW, Germany, 4 Center for Male Health, Stritch School of Medicine, Loyola University Medical Center, Maywood, IL, United States of America

⊙ These authors contributed equally to this work.
‡ KN, DG and SF also contributed equally to this work.
* sirikan.rojanasarot@bsci.com

**Data Availability Statement:** All relevant data are within the manuscript and its Supporting Information files. Specifically, adverse event rates

## Abstract

Treatment options for men with moderate-to-severe lower urinary tract symptoms (LUTS) due to benign prostatic hyperplasia (BPH) have variable efficacy, safety, and retreatment profiles, contributing to variations in patient quality of life and healthcare costs. This study examined the long-term cost-effectiveness of generic combination therapy (CT), prostatic urethral lift (PUL), water vapor thermal therapy (WVTT), photoselective vaporization of the prostate (PVP), and transurethral resection of the prostate (TURP) for the treatment of BPH. A systematic literature review was performed to identify clinical trials of CT, PUL, WVTT, PVP, and TURP that reported change in International Prostate Symptom Score (IPSS) for men with BPH and a prostate volume ≤80 cm³. A random-effects network meta-analysis was used to account for the differences in patient baseline clinical characteristics between trials. An Excel-based Markov model was developed with a cohort of males with a mean age of 63 and an average IPSS of 22 to assess the cost-effectiveness of these treatment options at 1 and 5 years from a US Medicare perspective. Procedural and adverse event (AE)-related costs were based on 2021 Medicare reimbursement rates. Total Medicare costs at 5 years were highest for PUL ($9,580), followed by generic CT ($8,223), TURP ($6,328), PVP ($6,152), and WVTT ($2,655). The total cost of PUL was driven by procedural ($7,258) and retreatment ($1,168) costs. At 5 years, CT and PUL were associated with fewer quality-adjusted life years (QALYs) than WVTT, PVP, and TURP. Compared to WVTT, the incremental cost-effectiveness ratios (ICERs) for both TURP and PVP were above a willingness-to-pay threshold of $50,000/QALY (TURP: $64,409/QALY; PVP: $87,483/QALY). This study provides long-term cost-effectiveness evidence for several common treatment options for men with BPH. WVTT is an effective and economically viable treatment in resource-constrained environments.

used in the model from different time points are included in S2 Table. Utility inputs are presented in S1 Table. All cost model inputs, including procedural, adverse event, retreatment and follow-up care costs, are in listed in Table 1. Total costs at 1 year and 5 years are summed of treatment, adverse events and follow up, and retreatment costs presented in Fig 2.

**Funding:** This study was funded by Boston Scientific. S Rojanasarot, B Cutone, and S Bhattacharyya are employees of Boston Scientific. K Neeser, D Gultyaev, and S Fu are employees of Certara Evidence & Access, a consulting company that is paid by Boston Scientific for the services rendered. A El-Arabi has no disclosures. Drs Chughtai, El-Arabi, and McVary were not compensated for their participation in this study.

**Competing interests:** B Chughtai is a paid clinical consultant for Boston Scientific, MedeonBio, Olympus, and Allergan as well as an investigator for Teleflex. S Rojanasarot, B Cutone, and S Bhattacharyya are employees of Boston Scientific. K Neeser, D Gultyaev, and S Fu are employees of Certara Evidence & Access. K McVary is a principal investigator for NIDDK and Urovant. He is a paid clinical consultant for Boston Scientific and MedeonBio. This does not alter our adherence to PLOS ONE policies on sharing data and materials.

## Introduction

Lower urinary tract symptoms (LUTS) attributed to benign prostatic hyperplasia (BPH) is a common chronic condition that negatively impacts patient health outcomes and results in substantial costs to the US healthcare system. Approximately 50–75% of men aged 50 years or older and 80% of men aged at least 70 years are diagnosed with LUTS-BPH [1]. A majority of these men experience LUTS, resulting in reduced quality of life (QOL) and increased risk of anxiety and depression [2, 3]. In 2013, Medicare was estimated to have spent more than $1.5 billion on office and outpatient services related to LUTS-BPH [4]. To contain the costs of treating LUTS-BPH, which will likely continue to rise as American society ages, it is critical to identify cost-effective treatment options that improve patient QOL and reduce the cost burden to payers.

Treatment options for men with moderate-to-severe LUTS-BPH range from medical combination therapy (CT) to minimally invasive surgical treatments (MISTs) to invasive surgical procedures [5, 6]. While CT is often prescribed to manage LUTS, the associated adverse events (AEs) and daily dosing regimens result in low patient adherence and the need for further LUTS-BPH retreatments [3]. Transurethral resection of the prostate (TURP) is considered the gold standard surgical treatment for men with moderate-to-severe LUTS-BPH [3]. TURP effectively removes obstructive prostate tissue but is also associated with long-term AEs such as erectile and ejaculatory dysfunction, and rarely urinary incontinence [6]. Compared to TURP, photoselective vaporization of the prostate (PVP) is a non-inferior alternative surgical procedure with shorter patient recovery times [6]. Although TURP and PVP are effective in improving symptoms, they are hospital-based procedures associated with long-term AEs affecting both healthcare costs and patient QOL.

Prostatic urethral lift (PUL) and water vapor thermal therapy (WVTT) are the newest MISTs with identical moderate recommendations by the American Urological Association (AUA) Guideline for men with BPH desiring to preserve sexual function [6]. PUL relieves LUTS by using permanent implants to compress obstructive prostatic tissues, while WVTT uses radiofrequency to generate water vapor that penetrates prostate tissue interstices, disrupts tissue cell membranes, and ablates tissue. These MISTs are effective in relieving LUTS and can be performed in non-hospital settings, including physician offices and ambulatory surgery centers (ASCs).

Previous cost-effectiveness analyses (CEAs) have addressed the cost-effectiveness of LUTS-BPH treatments, yet comprehensive analyses demonstrating the long-term cost-effectiveness of key treatment options for men with moderate-to-severe LUTS-BPH have not been conducted. A 2-year CEA of CT, MISTs, and invasive surgical procedures examined changes in International Prostate Symptom Score (IPSS) as the effectiveness outcome for each treatment option [7]. The analysis did not capture the QOL impact of the index treatments, AEs, or retreatments, which may have underestimated the effect of each BPH treatment on overall patient outcomes. Chughtai et al. focused on the long-term cost-effectiveness and budget impact of PUL compared to WVTT from a US Medicare perspective, but did not consider medical therapy or more invasive surgical procedures in the analysis [8]. These two studies have shown consistent results to the extent that WVTT represents the least expensive treatment option.

## Objectives

There is a need for short-term and long-term cost-effectiveness evidence supporting treatment options for men with moderate-to-severe LUTS-BPH. Such evidence would help practitioners and healthcare decision-makers select cost-effective treatment options for men with BPH in

resource-constrained environments. Therefore, the objective of this study was to estimate the cost-effectiveness of CT, PUL, WVTT, PVP, and TURP for men with moderate-to-severe BPH from a US Medicare perspective.

## Materials and methods

A CEA of CT, PUL, WVTT, PVP, and TURP was performed from a US Medicare perspective over a 5-year time horizon. An Excel-based (Microsoft, Redmond, WA) Markov model was developed using efficacy and safety data derived from a systematic literature review (SLR). To appropriately reflect both short-term and long-term changes in IPSS and AE rates, a 3-month cycle length was applied for the first year, followed by a 1-year cycle length for years 2 through 5. The model was populated with a cohort of males with a mean age of 63 years and an average IPSS of 22, using the baseline characteristics of patients in the Rezum II trial [9]. The trial characteristics were selected as these patients represent men with LUTS/BPH undergoing a treatment in the most recent trial among the included five treatment options. Cost-effectiveness was evaluated using a willingness-to-pay threshold of $50,000 per quality-adjusted life year (QALY) gained, a commonly used threshold for CEA, and presented as an incremental cost-effectiveness ratio (ICER) at years 1 and 5 [10]. A 3% annual discount rate was applied to both QALYs and costs, as recommended by the International Society for Pharmacoeconomics and Outcomes Research (ISPOR) and economic guidelines for US health economic analyses [11, 12]. Since this study does not involve human participants, neither institutional review board approval nor participant consent was required.

### Model structure and patient pathway

Men with moderate-to-severe LUTS/BPH entered the CEA model and were assigned to one of the five treatments (Fig 1). Patients who underwent CT received a fixed-dose combination of tamsulosin and dutasteride, a commonly prescribed combination for moderate-to-severe LUTS/BPH in the US [6]. Patients undergoing PUL, WVTT, PVP, or TURP could experience post-procedure catheterization. At each model cycle, all patients could either experience LUTS-related AEs, require retreatment, or receive follow-up care. The costs and utility weights of each cycle were accumulated yearly until year 5. Since LUTS/BPH is not considered a life-threatening condition, this analysis included only age-dependent mortality rates of US men based on the National Vital Statistics Reports [13].

### Clinical inputs

A SLR of randomized and non-randomized clinical trials of the five treatment options was conducted by systematically searching the Medline, EMBASE, Cochrane, and HTA databases. The search included full-text articles published between 2005 and 2020 and conference abstracts published between 2017 and 2020. The focus was on sources that reported changes in IPSS, post-procedure catheterization, retreatment rates, and LUTS-related AEs in men with moderate-to-severe LUTS/BPH and prostate volumes ≤80 cm$^3$ who were treated with either CT, PUL, WVTT, PVP, or TURP.

Following their index treatment, patients could experience changes in IPSS indicative of improvement or decline from baseline in each model cycle. We conducted a random-effects network meta-analysis (NMA) to obtain the adjusted IPSS changes for each treatment used in this CEA. For this analysis, the random-effects NMA model was combined with an aggregate regression model to conduct the indirect comparison. The baseline age and IPSS were used to account for the heterogeneity between trials and the difference in the distribution of baseline characteristics between comparators.

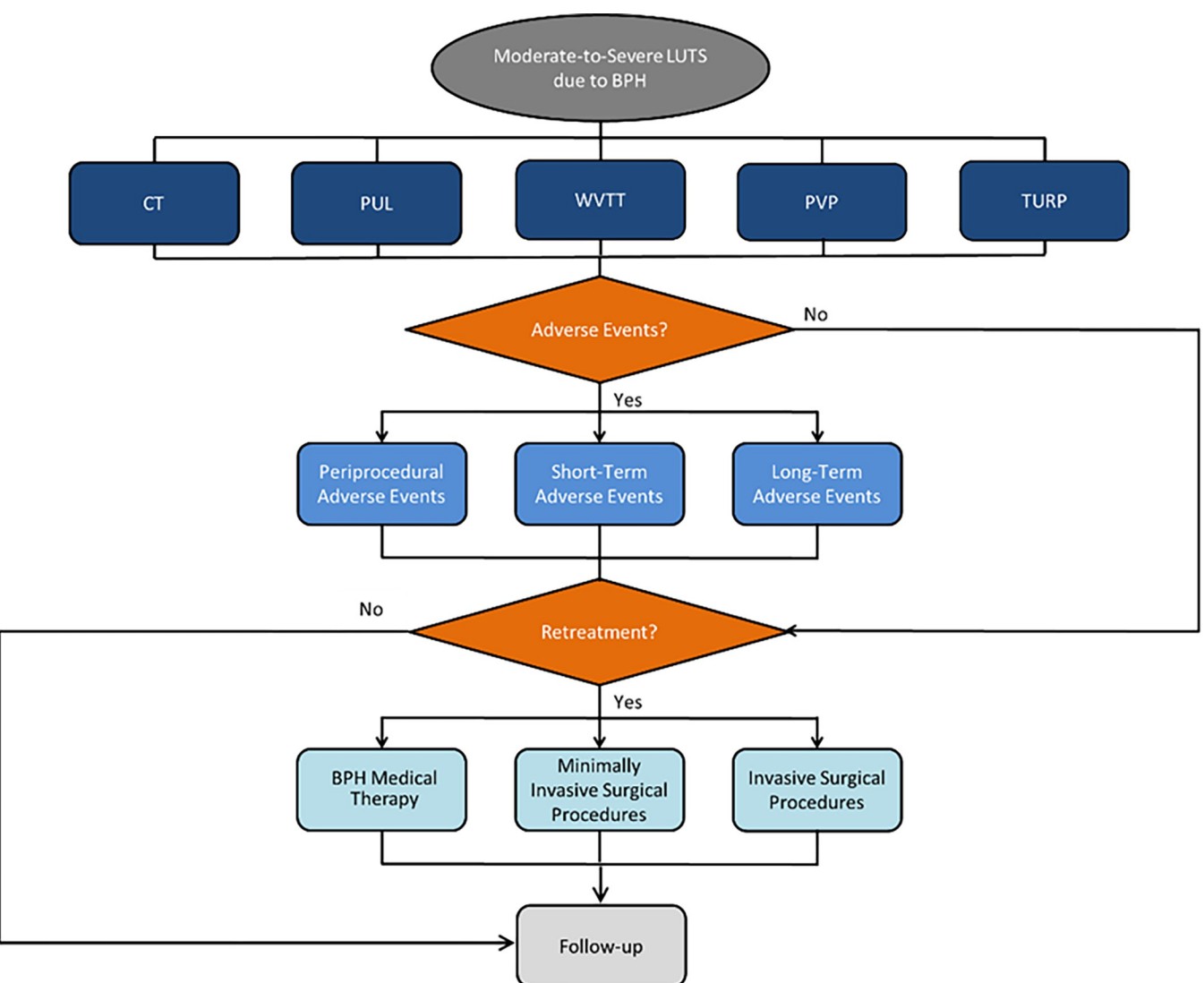

**Fig 1. Model schematic describing the patient pathway of the five treatment options for men with moderate-to-severe LUTS due to BPH.** Abbreviations: BPH, benign prostatic hyperplasia; CT, combination therapy; LUTS, lower urinary tract symptoms; PUL, prostatic urethral lift; PVP, photoselective vaporization of the prostate; TURP, transurethral resection of the prostate; WVTT, water vapor thermal therapy.

Post-procedure catheterization was assumed to occur at the time of the index surgical treatment, associated with a short-term disutility. The proportion of patients who underwent catheterization and the time from catheterization to catheter removal was derived from SLR. The catheterization rates were: 51.4% [14] for PUL, 90.4% [9] for WVTT, 94.1% [15] for PVP, and 93.2% [15] for TURP. The treatment-specific mean duration of catheterization is listed in S1 Table. It was assumed 50% of catheterized patients required an additional office visit for catheter removal.

Only LUTS-related AEs impacting QOL and healthcare costs were included in this analysis. AE rates were obtained from SLR and assigned to one of three categories (S2 Table): periprocedural AEs occurring at the same time as the index treatment (i.e., transurethral resection [TUR] syndrome, transfusions, immediate acute urinary retention [AUR]); short-term AEs resolving within 6 months (i.e., bladder spasm, urinary retention, urinary tract infection, pelvic

pain, hematuria, dysuria, urinary urge incontinence, frequency and urgency, encrusted implant, urethral stricture, bladder neck contraction); and long-term AEs lasting more than 6 months (i.e., erectile dysfunction, urinary incontinence). Since some AE rates were not available in the literature (S2 Table), we extrapolated the rates from 2 years to 5 years by applying the change of the AE rates between the second-last and last available quarter to subsequent cycles until the AE rates became 0%.

If an index treatment failed, retreatment would become necessary. Retreatment rates for each of the five treatment options were retrieved from their respective trials via SLR. Patients who experienced failed index CT treatment were retreated with MIST, reflecting a common BPH patient pathway. Patients who experienced failed MIST or surgical procedures could either receive medical therapy of generic tamsulosin 400 μg once daily to manage their recurring symptoms, undergo the same treatment again, or undergo more invasive surgical procedures. This assumption was used because the retreatment procedures for these patients could not be less invasive than their index treatments [16].

Patients who received CT as their index treatment required 1 follow-up office visit to receive the first prescription and another office visit within 1 month for follow-up. The same office visit requirement was also applied to patients who received medical therapy as their retreatment. From year 2 onward, CT patients required 1 office visit per year for potential dose adjustment. Patients with index surgical procedures were assumed to require 2 office visits within the first 3 months post-index procedure for follow-up care. Regarding annual follow-up care, 1 office visit per year was assigned to patients with mild and moderate LUTS, while 2 office visits per year were assigned to patients with severe LUTS. One post-void residual test per year was also assigned to patients with moderate and severe LUTS.

## Health state utilities

The health state-specific utilities and disutilities were used to calculate the patient QOL for each time point. Utility is a number between 0 (death) and 1 (perfect health) that represents the quality of a health state. On the other hand, disutility represents the decrement in utility resulting from a particular symptom or complication [17]. Thus, both utility and disutility are used to capture changes in QOL as a consequence of treatment. Utility values for LUTS/BPH severity levels and disutility values associated with post-procedure catheterization, LUTS-related AEs, and retreatments were derived from the published literature [18–21]. The Markov model used the mean and standard deviation (SD) of IPSS to estimate the proportion of patients with mild, moderate, or severe LUTS/BPH and mapped to the corresponding utility values for each of the LUTS/BPH severity levels. The disutility value of post-procedure catheterization was applied for a fixed time period using the mean duration of catheterization (S1 Table). The utility decrement per periprocedural, short-term, and long-term AE (S1 Table) was applied for the duration of the AE using the mean time to recovery retrieved from the Rezum II trial (unpublished data). The utility values were summed over the simulation period to obtain the overall QALYs.

## Cost inputs

Procedural, AE, retreatment, and follow-up care costs (Table 1) were based on 2021 Medicare reimbursement rates, which include a combination of facility and non-facility fees [22]. The procedural costs vary by site of service. Therefore, site of service proportions for each surgical procedure were obtained from the 5% Medicare Standard Analytic Files of procedures performed in 2019 and used to calculate weighted average procedural costs [23]. The procedural cost of PUL varies based upon the number of implants used; thus, this value was calculated

**Table 1. Weighted average procedure costs, follow-up costs, subsequent retreatment costs, periprocedural AE costs, short- and long-term AE costs.**

| | Cost | Code and Description |
|---|---|---|
| **Weighted average procedure costs** | | |
| CT | $3.73/day | Generic fixed-dose combination of tamsulosin and dutasteride [24, 25] |
| PUL | $7,258 | CPT 52441, 52442; HCPCS C9740 [22] |
| WVTT | $1,867 | CPT 53854 [22] |
| PVP | $4,813 | CPT 52648; DRG 714 [22] |
| TURP | $5,157 | CPT 52601; DRG 714 [22] |
| **Follow-up cost** | | |
| Mild BPH annual follow-up cost | $87 | CPT 99213 [22] |
| Moderate BPH annual follow-up cost | $97 | CPT 99213, 51798 [22] |
| Severe BPH annual follow-up cost | $184 | CPT 99213, 51798 [22] |
| **Subsequent retreatment cost** | | |
| WVTT | $1,867 | CPT 53854 [22] |
| PUL | $7,258 | CPT 52441, 52442; HCPCS C9740 [22] |
| TURP | $5,157 | CPT 52601; DRG 714 [22] |
| PVP | $4,813 | CPT 52648; DRG 714 [22] |
| Annual cost BPH medical therapy | $105 | Generic tamsulosin [24, 25] |
| **Periprocedural AEs** | | |
| TUR syndrome | $6,554 | DRG 714 [22] |
| Transfusions | $6,554 | DRG 714 [22] |
| Immediate acute urinary retention | $0 | No additional payment as this AE cost is included in the index treatment cost. |
| **Short-term AEs** | | |
| Bladder spasm | $87 | 1 office visit (CPT 99213) [22] |
| Urinary retention | $174 | 2 office visits (CPT 99213) [22] |
| Urinary tract infection | $87 | 1 office visit (CPT 99213) [22] |
| Pelvic pain | $87 | 1 office visit (CPT 99213) [22] |
| Hematuria | $87 | 1 office visit (CPT 99213) [22] |
| Dysuria | $87 | 1 office visit (CPT 99213) [22] |
| Urinary urge incontinence | $87 | 1 office visit (CPT 99213) [22] |
| Frequency and urgency | $87 | 1 office visit (CPT 99213) [22] |
| Encrusted implants | $3,285 | CPT 52318 [22] |
| Urethral strictures | $3,347 | CPT 52341 [22] |
| Bladder neck contraction | $3,179 | CPT 52450 [22] |
| **Long-term AEs** | | |
| Erectile dysfunction | $1,370 | Annual cost of erectile dysfunction management [25, 26] |
| Urinary incontinence | $1,354 | Annual cost of urinary incontinence management [25, 27] |

Abbreviations: AE, adverse event; BPH, benign prostatic hyperplasia; CT, combination therapy; NA, not assessed; PUL, prostatic urethral lift; PVP, photoselective vaporization of the prostate; TURP, transurethral resection of the prostate; WVTT, water vapor thermal therapy.

CPT® is a registered trademark of the American Medical Association.

using an average number of 4.9 implants obtained from the LIFT trial [14]. The capital costs of surgical procedures were not included in this analysis as Medicare does not reimburse capital systems separately. In the US, anesthesia costs are included in the procedural costs; therefore,

no additional costs of anesthesia were included. The treatment costs associated with AEs were estimated based on the number of office visits or procedures required to treat each specific event (Table 1). The costs of surgical retreatments were equal to the index treatment costs plus their subsequent AE costs. Medical therapy costs, including index CT and retreatment medication costs, were retrieved from 2019 Medicare Part D Drug Spending [24] and were inflated to 2021 dollars [25].

### Sensitivity analyses

This study utilized one-way sensitivity analyses (OWSA), probabilistic sensitivity analyses (PSA), and scenario analyses to assess the quantitative and qualitative impact of included parameters on model results. To identify the parameters that had the greatest impact on model results, OWSA was performed in which the input parameters were varied by ±10%. Tornado diagrams were constructed to depict the 10 most influential parameters on model results to estimate the degree of distribution of the calculated outcomes. We examined the level of confidence of the calculated QALYs and cost outcomes by conducting a PSA with 1,000 simulations, with each simulation representing one patient. For each simulation, the input values were randomly varied within the lower and upper limits of the SD of each parameter. PSA results were presented as scatterplots. Since this analysis involved NMA, a scenario analysis was conducted using IPSS values derived from a fixed-effects NMA model to compare the model results with the results of the random-effects NMA.

### Results

The SLR identified 3,014 abstracts that were screened to fulfil the predefined inclusion criteria, including study type, study population, index treatment, and outcomes. A total of 237 abstracts were included for full-text review. Of these, 20 publications (16 randomized controlled trials, 4 non-randomized trials; S3 Table) were included in the NMA: 15 publications including 9 treatments for the IPSS change at 1 year and 5 publications including 7 treatments for the retreatment rate at 1 year. In addition, 12 publications (7 randomized controlled trials, 5 non-randomized trials) provided information about periprocedural, LUTS-related short- and long-term AE rates, and utility values used in the model. The baseline prostate volumes of patients in the 20 included publications ranged from 33.1 cm$^3$ to 80.0 cm$^3$ while their baseline peak flow rates ranged from 4.5 mL/s to 9.9 mL/s.

The random-effects NMA model showed (Table 2) that at 1 year, BPH patients with a mean age of 63 years and a baseline IPSS score of 22 who were treated with invasive surgical procedures experienced the greatest IPSS improvement (-Δ14.1 for TURP and -Δ13.8 for PVP). Among non-invasive treatments, WVTT had the greatest 1-year IPSS improvement (-Δ11.7) while the IPSS improvement for PUL and CT was similar (-Δ10.4 vs -Δ10.3). The NMA revealed that WVTT was associated with the lowest retreatment rate at 1 year (3.0%), followed by CT (3.6%), TURP (6.3%), PVP (7.8%), and PUL (8.0%).

Fig 2 reports the total costs of all five treatments at years 1 and 5, ranging from the least expensive treatment (WVTT) to most expensive treatment (PUL). The total Medicare costs associated with CT and WVTT were similar at 1 year ($2,194 and $2,019, respectively). At 5 years, the total costs of CT and WVTT differed substantially ($8,223 and $2,655, respectively). When comparing the total costs of the four surgical procedures at 5 years, PUL was 1.5 times more expensive than PVP and TURP and 3.5 times more expensive than WVTT (Table 3).

For each of the five treatment options, procedural costs were the major cost component of the total costs. Because the procedural costs associated with surgical treatments are accounted for in the first year, the costs associated with surgical treatments accrued in subsequent years

**Table 2. IPSS change from baseline to year 5 from the random-effects network meta-analysis model for use in the base-case cost-effectiveness analyses.**

|  | CT | | PUL | | WVTT | | PVP | | TURP | |
|---|---|---|---|---|---|---|---|---|---|---|
|  | Mean | ±SD | Mean | ±SD | Mean | ±SD | Mean | ±SD | Mean | ±SD |
| Baseline | 22.0 | 4.8 | 22.0 | 4.8 | 22.0 | 4.8 | 22.0 | 4.8 | 22.0 | 4.8 |
| 3 months | 11.0 | 6.4 | 10.9 | 6.4 | 10.6 | 6.4 | 10.1 | 6.4 | 10.0 | 6.4 |
| 6 months | 10.5 | 6.2 | 10.5 | 6.2 | 9.8 | 6.2 | 8.7 | 6.2 | 8.6 | 6.2 |
| 1 year | 11.7 | 7.0 | 11.6 | 7.0 | 10.3 | 6.7 | 8.2 | 7.3 | 7.9 | 7.2 |
| 2 years | 11.6 | 6.5 | 11.5 | 6.5 | 10.2 | 6.2 | 8.1 | 6.8 | 7.8 | 6.7 |
| 3 years | 11.9 | 6.4 | 11.8 | 6.4 | 10.5 | 6.1 | 8.4 | 6.7 | 8.1 | 6.6 |
| 4 years | 12.8 | 7.7 | 12.7 | 7.7 | 11.4 | 7.4 | 9.3 | 8.0 | 9.0 | 7.9 |
| 5 years | 12.5 | 8.1 | 12.4 | 8.1 | 11.1 | 7.8 | 9.0 | 8.4 | 8.7 | 8.3 |

Abbreviations: CT, combination therapy; IPSS, International Prostate Symptom Score; PUL, prostatic urethral lift; PVP, photoselective vaporization of the prostate; SD, standard deviation; TURP, transurethral resection of the prostate; WVTT, water vapor thermal therapy.

The analysis was conducted using an average IPSS of 22 as a baseline for all treatments. The IPSS changes of WVTT from years 1 to 5 were derived from 5-year Rezum II trial results [28]. The difference of the IPSS changes between WVTT and PUL, WVTT, PVP, and TURP, respectively, were derived from the NMA results and applied to each time point.

are attributable to AEs, follow-up care, and retreatment. In contrast, CT, which is an ongoing therapy, accumulates treatment costs constantly over time. The highest procedural costs were associated with PUL, 41% higher than those of TURP, the next most expensive procedure. Regarding retreatment, TURP had the lowest retreatment cost at 5 years followed by PVP, WVTT, CT, and PUL (Fig 2).

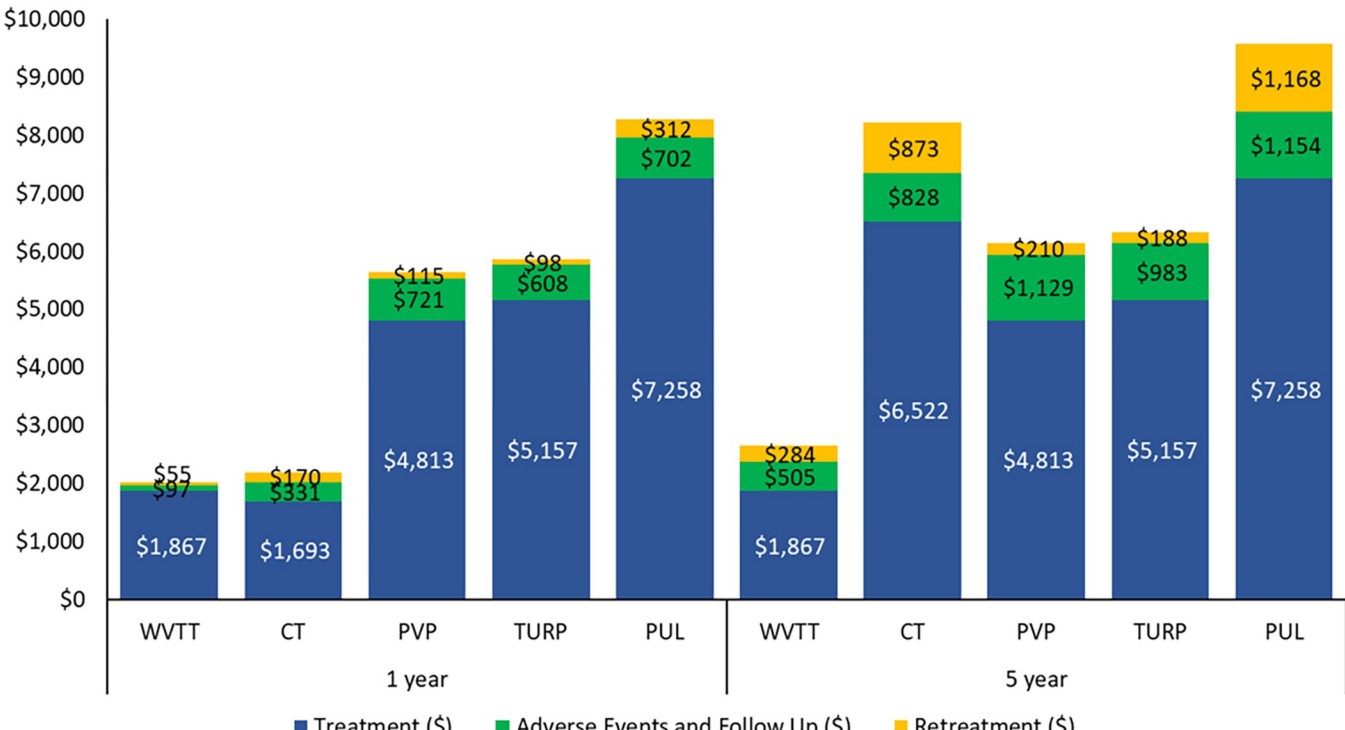

**Fig 2. Medicare per patient costs at year 1 and year 5 for the five treatment options for men with moderate-to-severe lower urinary tract symptoms due to benign prostatic hyperplasia ranging from the least to the most expensive at year 1.** Abbreviations: CT, combination therapy; PUL, prostatic urethral lift; PVP, photoselective vaporization of the prostate; TURP, transurethral resection of the prostate; WVTT, water vapor thermal therapy.

**Table 3. Costs, QALYs, and ICER at 5 years for the five treatment options for men with moderate-to-severe lower urinary tract symptoms due to benign prostatic hyperplasia.**

|  | Generic CT | PUL | WVTT | PVP | TURP |
|---|---|---|---|---|---|
| Total cost | $8,223 | $9,580 | $2,655 | $6,152 | $6,328 |
| Total QALYs | 4.118 | 4.141 | 4.189 | 4.229 | 4.246 |
| Life years | 4.799 | | | | |
| Incremental cost relative to CT | --- | $1,357 | -$5,567 | -$2,071 | -$1,895 |
| Incremental QALYs relative to CT | --- | 0.023 | 0.071 | 0.111 | 0.128 |
| ICER versus CT | --- | $57,888/QALY | dominates | dominates | dominates |

Abbreviations: CT, combination therapy; ICER, incremental cost-effectiveness ratio; PUL, prostatic urethral lift; PVP, photoselective vaporization of the prostate; QALYs, quality-adjusted life years; TURP, transurethral resection of the prostate; WVTT, water vapor thermal therapy.

Total costs were rounded to whole dollars and total QALYs were rounded to 3 decimal points. The exact total cost and QALY values were used to calculate all reported ICERs.

In this study, QALYs were driven by the treatment-specific efficacy, AE rates, and retreatment rates, given no treatment-related mortality was associated with any of the five treatments. Among non-invasive treatments, WVTT showed the highest QALYs compared to CT and PUL, while TURP and PVP had similar QALYs at 5 years (Table 3). Taking into account the total costs and QALYs associated with each treatment option, CT was found to be both more expensive and less effective than TURP and WVTT over the 5-year time horizon (Table 3). The ICERs for both TURP and PVP compared to WVTT were above a willingness-to-pay threshold of $50,000/QALY (TURP: $64,409/QALY; PVP: $87,483/QALY).

## Sensitivity analyses

**Probabilistic sensitivity analysis.** Results of the PSA simulations (Fig 3A) indicate compared with TURP, CT was more expensive and less effective in 98% and 85% of the 1,000 PSA simulations, respectively. A similar result was seen in the comparison of PVP to CT (Fig 3B) with higher costs in 99% and lower QALYs in 85% for CT. The comparison of PUL to CT demonstrated that PUL is more expensive than CT (92% of all simulations) and provides higher QALYs in only 63% of all PSA simulations (Fig 3C). PSA of CT versus WVTT indicate WVTT is always less costly than CT (100% of simulations) and associated with higher QALYs than CT in 89% of all simulations (Fig 3D).

**One-way sensitivity analysis.** For OWSA, CT was used as the reference therapy given it is the least effective treatment and the only non-surgical treatment considered in this study. OWSA has shown for the comparison of TURP vs CT (S1A Fig) and PVP vs CT (S1B Fig) that the parameters "Moderate BPH Utility," "CT Treatment Cost," "TURP Treatment Cost," and "PVP Treatment Cost," respectively, had the highest impact on the model results. The OWSA of PUL vs CT (S1C Fig), revealed "PUL Treatment Cost," "CT Treatment Cost," and "Cost Discount" to be the most impactful parameters on the ICER. The most influential parameters in the comparison of WVTT vs CT (S1D Fig) were: "Severe BPH Utility," "CT Treatment Cost," and "Moderate BPH Utility."

**Scenario analysis.** The results of the fixed-effects model are presented in S4 Table. This scenario utilized the IPSS values derived from the fixed-effects NMA model (S5 Table). The 5-year overall costs based on the IPSS scores from the random-effects model (Table 3) for PUL, PVP, and TURP were slightly lower than the costs based on the IPSS values derived from the fixed-effects model. The QALYs in the random-effects model were higher for CT, PUL, PVP, and TURP than in the fixed-effects model. The total cost and QALYs of WVTT were

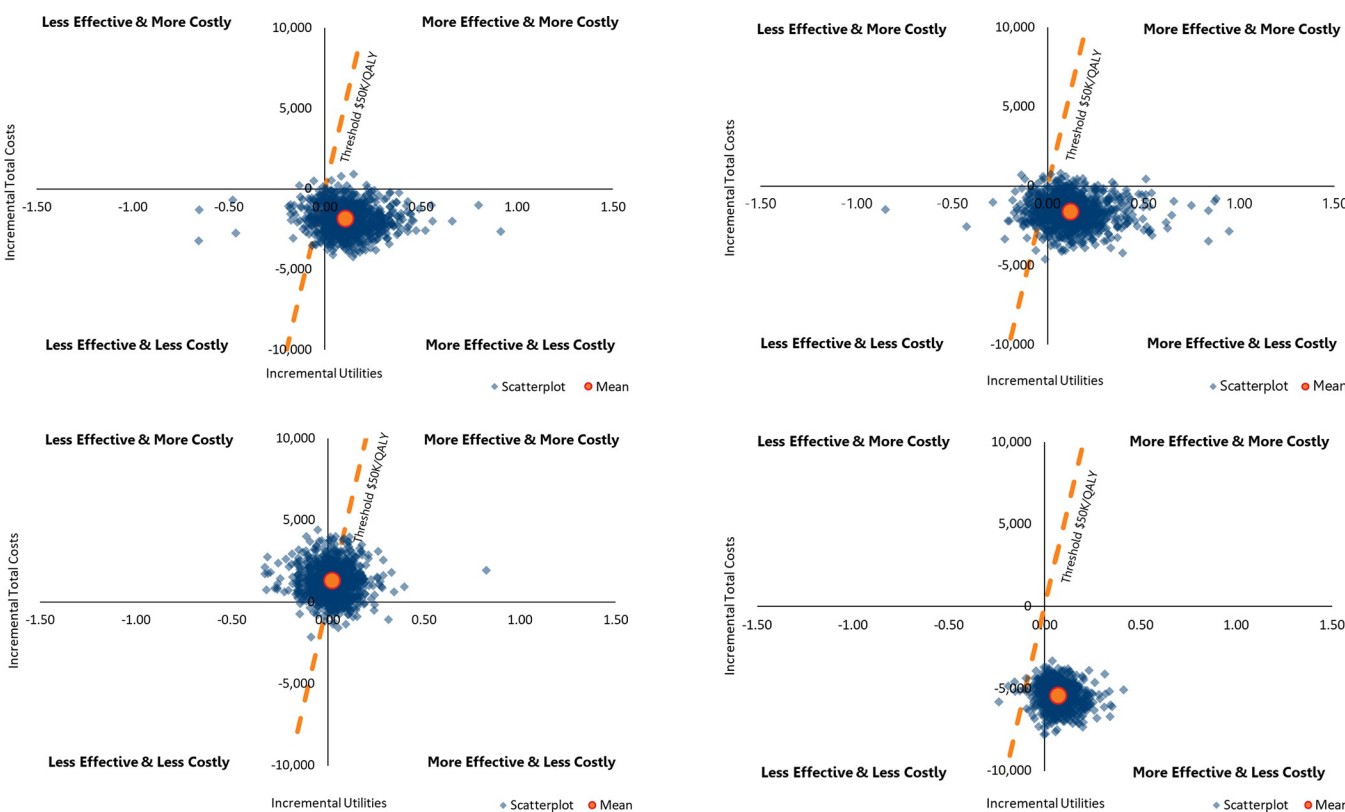

**Fig 3. Scatterplots based on PSAs by comparing CT versus surgical treatments. a**: TURP vs. CT at 5 years. **b**: PVP vs. CT at 5 years. **c**: PUL vs. CT at 5 years. **d**: WVTT vs. CT at 5 years. Abbreviations: CT, combination therapy; PSA, probabilistic sensitivity analysis; PUL, prostatic urethral lift; PVP, photoselective vaporization of the prostate; QALYs, quality-adjusted life years; TURP, transurethral resection of the prostate; WVTT, water vapor thermal therapy.

equal for both NMA models (reference treatment in the NMA). Using the fixed-effects model, the ICER for PUL changed from $57,888/QALY to $126,619/QALY if compared to CT, while the ICERs of the other comparisons remained dominant.

## Discussion

This study evaluated the clinical and cost-effectiveness outcomes of CT, PUL, WVTT, PVP, and TURP for men experiencing moderate-to-severe LUTS due to BPH and a prostate volume up to 80 cm$^3$. A random-effects NMA found that TURP and PVP, the two invasive surgical procedures considered in this analysis, resulted in the greatest IPSS improvement, followed by WVTT, PUL, and CT. The highest total costs at 5 years were seen in PUL, followed by CT, TURP, PVP, and WVTT. TURP and PVP were associated with higher QALYs, but at a substantially higher cost than WVTT. From an economic perspective, CT was more costly and less effective than TURP and WVTT, with WVTT being the most cost-effective therapy. Given these results, TURP and PVP remain useful as more invasive surgical treatment options and could be considered during shared decision processes. Due to the significant costs and modest clinical benefits associated with CT, WVTT has the potential to be considered as an early therapy for men with moderate-to-severe LUTS/BPH, given its economic and clinical profile. In a resource-constrained environment, WVTT appears to be an economically viable treatment option for men with LUTS/BPH.

Compared to a fixed-dose combination of tamsulosin and dutasteride, PUL, and invasive surgical procedures, WVTT was a considerably less expensive treatment option from years 1

to 5, while the QALYs associated with WVTT were in between those of the remaining four treatment options. WVTT is a less costly alternative to the other treatment options at year 5 mainly due to the low procedural costs, high efficacy of the treatment, and low AE rates.

This study found that TURP is associated with a greater IPSS improvement following treatment when compared to other treatment alternatives, such as MISTs. However, the invasiveness of TURP was also associated with an increased risk of short-term and long-term AEs [3]. According to the latest version of the AUA Guidelines for the surgical management of LUTS/ BPH, men should be informed about the sexual AEs of BPH surgery [6]. Given LUTS/BPH management is centered on shared decision-making between patients and their urologists, alternative surgical procedures that preserve erectile and ejaculatory function should be discussed with patients. These same Guidelines also recommended that men with LUTS/BPH desiring to preserve their erectile and ejaculatory function should be offered WVTT or PUL. The present CEA found the total costs of PUL were 3 times higher than those of WVTT and that PUL was associated with lower QALYs than PUL. This study used 4.9 implants as reported in the LIFT trial [14] to calculate the procedural cost of PUL, and the cost depends on the number of implants. As a result, the cost difference between PUL and WVTT could be even higher if the maximum reimbursable implant numbers allowed by Medicare were used or if a higher proportion of PUL was performed in an ASC setting. This economic finding provides direction for our health policy such that WVTT should be noted to offer a more financially viable option.

Although there have been previous health economic analyses of treatment options for LUTS/BPH, they have primarily focused on clinical outcome improvement and short-term cost-effectiveness analyses. The present study is the first US-based CEA that considers long-term outcomes of medical therapy, MISTs, as well as more invasive surgical procedures. Ulchaker et al. conducted a CEA of CT, PUL, WVTT, conductive radiofrequency thermal therapy, PVP, and TURP [7]. The study found the greatest improvements in IPSS were achieved by TURP and PVP, and among the MISTs studied, WVTT resulted in a greater IPSS improvement than PUL. The IPSS improvement findings from Ulchaker et al. are in line with the results from this present study. Ulchaker et al. reported the economic outcomes over a period of 2 years, whereas the present analysis examined both short-term and long-term economic outcomes at years 1 and 5. Ulchaker et al. found CT was associated with the lowest total costs at 2 years, followed by WVTT, PVP, TURP, and PUL. This varies from the present study in which WVTT was the least costly treatment option from year 1 to year 5. The differences in results between these two studies emphasize the importance of using a longer time horizon in economic analyses to take into account all effects that may occur over time. Despite a few deviations between the approach taken by Ulchaker et al. and the approach used in our analysis, which used QALYs as an effectiveness outcome, focused on a longer time horizon, and incorporated NMA, it can be concluded both economic analyses point in the same overall direction in terms of treatment effectiveness and ICER.

This study is the first comprehensive CEA that reports the long-term cost-effectiveness of the commonly used BPH treatment options in the US ranging from CT, MIST, and surgical treatments over a 5-year time horizon. The results of this study could be used to inform decision-makers, including health technology assessment bodies outside of the US, when they assess the health economic value of each treatment option and adjust or establish reimbursed prices to ensure the balance between budget sustainability and patient access. Additionally, the present study incorporated an SLR and an NMA that provided clinical insights and solid clinical inputs for the model, included both random-effects and fixed-effects NMAs and found consistent results, and completed several sensitivity analyses to ensure the reliability of the model results. A majority of the sensitivity analyses showed the outcomes to be stable over a

wide range of assumptions, demonstrating the reliability of model findings. Given the reliable results, it is likely savings through the use of WVTT will apply to commercially insured men since private insurers normally reimburse providers at higher reimbursement rates.

This analysis has some limitations. First, like other CEAs, a portion of the clinical and safety inputs used in the CEA model were derived from controlled trials. The characteristics of patients in the trials may vary from patients seen in real-life practice. The model inputs were verified by a medical expert practicing in the US setting to minimize the impact of this issue and to ensure the applicability of the model results to men with LUTS/BPH in the US. Second, while NMAs were conducted, the analyses of IPSS improvements were limited to 1 year due to the unavailability of long-term IPSS data across all treatment options. Thus, targeted searches were performed to close the data gap for the remaining 4 years. Last, this model used a US Medicare perspective, so the model was structured using US reimbursement and cost structures. Due to country-specific differences in health care systems, future research could focus on developing a CEA using different perspectives to confirm the cost-effectiveness findings of WVTT for men with moderate-to-severe LUTS/BPH in other countries. Nevertheless, as shown in various sensitivity analyses, the model results of this present study were not sensitive to the changes in model inputs. Therefore, this study could provide evidence to support decision-making processes by health technology assessment authorities outside of the US.

This CEA provided clinical and cost-effectiveness evidence of five treatment options for LUTS due to BPH. Since the focus of the medical field has centered on evidence-based practices, the present study can equip practitioners and decision-makers with comprehensive clinical and cost-effectiveness findings. Therefore, this study could be used for patient-centered treatment consideration.

## Conclusion

This analysis showed that WVTT, over a short- and long-term time horizon, may be a suitable clinical and economic early therapy for eligible men with BPH. While TURP and PVP were associated with higher QALYs, these more invasive procedures have a substantially higher cost than WVTT. Compared to WVTT, CT and PUL are associated with lesser IPSS improvement and higher costs, which should be considered if these two treatments are selected.

## Supporting information

**S1 Table. Utility inputs.** Abbreviations: AE, adverse event; BPH, benign prostatic hyperplasia; CT, combination therapy; IPSS, International Prostate Symptom Score; NA, not applicable; PUL, prostatic urethral lift; PVP, photoselective vaporization of the prostate; TURP, transurethral resection of the prostate; WVTT, water vapor thermal therapy. #The utility score of PVP is not available. Since TURP and PVP are invasive surgical procedures, the utility score of TURP was applied for PVP. *The mean time to recovery assumptions were validated by the clinical expert as the data were not available in the literature. **The mean time to recovery of the AEs was retrieved from the Rezum II trial.
(DOCX)

**S2 Table. Time-specific AE rates.** Abbreviations: AE, adverse event; CT, combination therapy; PUL, prostatic urethral lift; PVP, photoselective vaporization of the prostate; TURP, transurethral resection of the prostate; WVTT, water vapor thermal therapy. *No AE was identified from published literature; 0.0% was assumed. †AE rates varied in the time frame. The lowest and highest AE rates were listed. ‡The inputs were extrapolated from 2 years to 5 years.
(DOCX)

**S3 Table. Bibliography of 20 publications used in the network meta-analysis.**
(DOCX)

**S4 Table. Costs, QALYs, and ICER at 5 years for the five treatment options for men with moderate-to-severe lower urinary tract symptoms due to benign prostatic hyperplasia derived from the fixed-effects model.** Abbreviations: CT, combination therapy; ICER, incremental cost-effectiveness ratio; PUL, prostatic urethral lift; PVP, photoselective vaporization of the prostate; QALYs, quality-adjusted life years; TURP, transurethral resection of the prostate; WVTT, water vapor thermal therapy. Total costs were rounded to whole dollars and total QALYs were rounded to 3 decimal points. The exact total cost and QALY values were used to calculate all reported ICERs.
(DOCX)

**S5 Table. IPSS change from baseline to year 5 from the fixed-effects network meta-analysis model for use in the sensitivity analysis.** Abbreviations: CT, combination therapy; IPSS, International Prostate Symptom Score; PUL, prostatic urethral lift; PVP, photoselective vaporization of the prostate; SD, standard deviation; TURP, transurethral resection of the prostate; WVTT, water vapor thermal therapy. The analysis was conducted using an average IPSS of 22 as a baseline for all treatments. The IPSS changes of WVTT from years 1 to 5 were derived from 5-year Rezum II trial results [1]. The difference of the IPSS changes between WVTT and PUL, WVTT, PVP, and TURP, respectively, were derived from the network meta-analysis results and applied to each time point.
(DOCX)

**S1 Fig. Tornado diagram of one-way sensitivity analysis at 5 years. a.** TURP vs CT. **b.** PVP vs CT. **c.** PUL vs CT. **d.** WVTT vs CT. Abbreviations: BPH, benign prostatic hyperplasia; CT, combination therapy; PUL, prostatic urethral lift; PVP, photoselective vaporization of the prostate; TURP, transurethral resection of the prostate; WVTT, water vapor thermal therapy.
(ZIP)

## Acknowledgments

The authors would like to thank Alysha McGovern for her proofreading assistance of this manuscript.

## Author Contributions

**Conceptualization:** Bilal Chughtai, Sirikan Rojanasarot, Kurt Neeser, Samir K. Bhattacharyya, Ahmad M. El-Arabi, Ben J. Cutone, Kevin T. McVary.

**Data curation:** Sirikan Rojanasarot, Kurt Neeser, Shuai Fu.

**Formal analysis:** Kurt Neeser, Dmitry Gultyaev, Shuai Fu.

**Funding acquisition:** Sirikan Rojanasarot, Samir K. Bhattacharyya.

**Investigation:** Sirikan Rojanasarot, Kurt Neeser, Dmitry Gultyaev, Shuai Fu.

**Methodology:** Bilal Chughtai, Sirikan Rojanasarot, Kurt Neeser, Dmitry Gultyaev, Shuai Fu, Kevin T. McVary.

**Project administration:** Sirikan Rojanasarot.

**Resources:** Bilal Chughtai, Sirikan Rojanasarot, Kurt Neeser, Kevin T. McVary.

**Software:** Dmitry Gultyaev, Shuai Fu.

**Supervision:** Sirikan Rojanasarot, Kurt Neeser.

**Validation:** Bilal Chughtai, Sirikan Rojanasarot, Kurt Neeser, Dmitry Gultyaev, Shuai Fu, Samir K. Bhattacharyya, Ahmad M. El-Arabi, Ben J. Cutone.

**Visualization:** Bilal Chughtai, Sirikan Rojanasarot, Kurt Neeser, Kevin T. McVary.

**Writing – original draft:** Sirikan Rojanasarot, Kurt Neeser.

**Writing – review & editing:** Bilal Chughtai, Sirikan Rojanasarot, Kurt Neeser, Dmitry Gultyaev, Shuai Fu, Samir K. Bhattacharyya, Ahmad M. El-Arabi, Ben J. Cutone, Kevin T. McVary.

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
