## [Decision Letter · Decision Letter 0]

11 Nov 2021

PONE-D-21-30773A comprehensive analysis of clinical, quality of life, and cost-effectiveness outcomes of key treatment options for benign prostatic hyperplasiaPLOS ONE

Dear Dr. Rojanasarot,

Thank you for submitting your manuscript to PLOS ONE. After careful consideration, we feel that it has merit but does not fully meet PLOS ONE’s publication criteria as it currently stands. Therefore, we invite you to submit a revised version of the manuscript that addresses the points raised during the review process.

  In your M&M you explain that you use the Rezum patient of ref 10 and the ‘model patient’. These patients had prostates 30-80gm and flowrates <15mL/s. I have two questions about that: can you describe this model patient including the objective parameters that I have mentioned to provide the reader with more clarity, but more important: I think that many of the studies that you referred to have also included persons with (much) smaller prostates. The minimum of 30g has a good reason (with TURP as the comparator) because we do know that (TURP)resection/‘desobstruction’ in patients with small volumes hardly improves flowrates. Can you speculate/or conclude or comment on how this has affected costs. Is lowering the threshold to surgery which may be a consequence related to this cost-effective? (and are interventions not relatively more prone to re-intervention when done for patients without BPH and or with (only) other dysfunctions than bladder outflow obstruction? Did you include (costs of) continued (LUT) medication in long term AE (not explicitly mentioned with ‘retreatment’ and or in the tables (these) mention a.e. symptoms but not treatment.

This also brings me to your first sentence ‘Lower urinary tract symptoms (LUTS) attributed to benign prostatic hyperplasia (BPH)…’ the word ‘attributed to’ is not particularly cost saving (for all types of management including medication), or am I wrong? I ask you to consider commenting on this.

Last: You extrapolated follow up in your model. Can you better explain in the main text P5L94-96 what you did?

We look forward to receiving your revised manuscript.

Kind regards,

Peter F.W.M. Rosier, M.D. PhD

Academic Editor

PLOS ONE

Journal Requirements:

[B Chughtai is a paid clinical consultant for Boston Scientific, MedeonBio, Olympus, and Allergan as well as an investigator for Teleflex. S Rojanasarot, B Cutone, and S Bhattacharyya are employees at Boston Scientific. K Neeser, D Gultyaev, and S Fu are employees of Certara Evidence & Access, a consulting company that is paid by Boston Scientific for the services render. A El-Arabi has no disclosures. K McVary is a principal investigator for NxThera, NIDDK, Astellas, Boston Scientific, and Olympus. Drs Chughtai, El-Arabi, and McVary were not compensated for their participation in this study.] 

Reviewers' comments:

Reviewer's Responses to Questions

**Comments to the Author**

1. Is the manuscript technically sound, and do the data support the conclusions?

Reviewer #1: Yes

Reviewer #2: Yes

2. Has the statistical analysis been performed appropriately and rigorously? 

Reviewer #1: Yes

Reviewer #2: Yes

3. Have the authors made all data underlying the findings in their manuscript fully available?

Reviewer #1: Yes

Reviewer #2: Yes

4. Is the manuscript presented in an intelligible fashion and written in standard English?

Reviewer #1: Yes

Reviewer #2: Yes

5. Review Comments to the Author

Reviewer #1: Minimal Invasive Techniques in benign prostatic hyperplasia increase the interest in these new procedures. Congratulation to the authors to focus on the expenses of these new techniques and the quality of life of the patients.

Reviewer #2: Well conducted comprehensive analysis. Highlights key differences between various mist procedures in accurate and technically sound fashion. Accounts for upfront and retreatment costs. It would be interesting to make note if claims data varies significantly from pivots trial data

6. PLOS authors have the option to publish the peer review history of their article (what does this mean?). If published, this will include your full peer review and any attached files.

Reviewer #1: No

Reviewer #2: No

---

## [Author Response · Author response to Decision Letter 0]

23 Nov 2021

Reviewers’ Comments 

In your M&M you explain that you use the Rezum patient of ref 10 and the ‘model patient’. These patients had prostates 30-80gm and flowrates <15mL/s. I have two questions about that: 

• Can you describe this model patient including the objective parameters that I have mentioned to provide the reader with more clarity, but more important: I think that many of the studies that you referred to have also included persons with (much) smaller prostates. The minimum of 30g has a good reason (with TURP as the comparator) because we do know that (TURP)resection/‘desobstruction’ in patients with small volumes hardly improves flowrates. 

Response: Thank you very much for the great clinical insight. In our systematic review and meta-analysis, we used the inclusion criteria of men with prostate volumes ≤80 cm3. The criteria are documented on page 7 of the manuscript. From the systematic review, we found that the baseline mean prostate volumes ranged from 33.1 cm3 to 80.0 cm3 while baseline peak flow rates ranged from 4.5 mL/s to 9.9 mL/s. To clarify this point, we included the following statement in the result section of the manuscript on page 14 “The baseline prostate volumes of patients in the 20 included publications ranged from 33.1 cm3 to 80.0 cm3 while their baseline peak flow rates ranged from 4.5 mL/s to 9.9 mL/s.” 

• Can you speculate/or conclude or comment on how this has affected costs. Is lowering the threshold to surgery which may be a consequence related to this cost-effective? (and are interventions not relatively more prone to re-intervention when done for patients without BPH and or with (only) other dysfunctions than bladder outflow obstruction? 

Response: This is a really good observation. The retreatment rates are one of the key components that drive the total treatment costs. If lowering the threshold to surgery reduces the retreatment rates, changing the criteria for surgery would have an impact on the cost-effectiveness results. For BPH management, clinical guidelines including the 2021 American Urological Association (AUA) Guideline for BPH1 and the indication for use of treatment options include specific prostate volume details. The prostate volume details determine available treatment options for men with BPH. Therefore, lowering the threshold to surgery may not adhere to recommended clinical guidelines. 

• Did you include (costs of) continued (LUT) medication in long term AE (not explicitly mentioned with ‘retreatment’ and or in the tables (these) mention a.e. symptoms but not treatment. 

Response: For men who had recurring LUTS due to BPH symptoms post-MIST or surgical procedures, they could receive BPH medical therapy of generic tamsulosin 40 mg once daily long term. We have previously included the information in Table 1 on page 10. To better clarify this point in the manuscript, we have updated the following statement in the clinical input section of the manuscript on page 8 “Patients who experienced failed MIST or surgical procedures could either receive medical therapy of generic tamsulosin 40 mg once daily to manage their recurring symptoms, undergo the same treatment again, or undergo more invasive surgical procedures.”

This also brings me to your first sentence ‘Lower urinary tract symptoms (LUTS) attributed to benign prostatic hyperplasia (BPH)…’ the word ‘attributed to’ is not particularly cost saving (for all types of management including medication), or am I wrong? I ask you to consider commenting on this. 

Response: “Lower urinary tract symptoms (LUTS) attributed to benign prostatic hyperplasia” is a common term used in the Urology field to describe the bothersome symptoms from an enlarged prostate gland. The same term is used in the AUA Guideline for the management of benign prostatic hyperplasia/ lower urinary tract symptoms.1 In this cost-effectiveness analysis context, only men with moderate-to-severe LUTS were considered given that if the conditions are not treated, the quality of life of these men would be negatively impacted. Men with anatomic enlargement of the prostate but without LUTS are not included as their first disease management is not a treatment but watchful waiting. 

Last: You extrapolated follow up in your model. Can you better explain in the main text P5L94-96 what you did? 

Response: Thank you for this comment. To clarify this point in the manuscript, we included the following statement in the clinical inputs section of the manuscript on page 8 “Since some AE rates were not available in the literature (S2 Table), we extrapolated the rates from 2 years to 5 years by applying the change of the AE rates between the second-last and last available quarter to subsequent cycles until the AE rates became 0%.” 

Reference:

1. Lerner LB, McVary, KT, Barry MJ et al: Management of lower urinary tract symptoms attributed to benign prostatic hyperplasia: AUA Guideline part II, surgical evaluation and treatment. J Urol 2021; 206: 818.

---

## [Decision Letter · Decision Letter 1]

18 Jan 2022

PONE-D-21-30773R1A comprehensive analysis of clinical, quality of life, and cost-effectiveness outcomes of key treatment options for benign prostatic hyperplasiaPLOS ONE

Dear Dr. Rojanasarot,

Thank you for submitting your manuscript to PLOS ONE. After careful consideration, we feel that it has merit but does not fully meet PLOS ONE’s publication criteria as it currently stands. Therefore, we invite you to submit a revised version of the manuscript that addresses the points raised during the review process.

ACADEMIC EDITOR: Few small comments to answer and or include in the manuscript.

We look forward to receiving your revised manuscript.

Kind regards,

Peter F.W.M. Rosier, M.D. PhD

Academic Editor

PLOS ONE

Journal Requirements:

Additional Editor Comments (if provided):

Thank you for your revision and answers to the question. You have however not explained why ref 39 contradict your statement regarding prostate sizes. You have added 'tamulosin 40mg (which should be tamsilosin 40μg) but in fact patients receive 'CT' again (that should also include antimuscarinics). This should be included (likely with more than just one 'office visit' in 5y) unless you have good arguments not to include this in the calculations.

Reviewers' comments:

Reviewer's Responses to Questions

**Comments to the Author**

1. If the authors have adequately addressed your comments raised in a previous round of review and you feel that this manuscript is now acceptable for publication, you may indicate that here to bypass the “Comments to the Author” section, enter your conflict of interest statement in the “Confidential to Editor” section, and submit your "Accept" recommendation.

Reviewer #1: All comments have been addressed

Reviewer #2: All comments have been addressed

2. Is the manuscript technically sound, and do the data support the conclusions?

Reviewer #1: Yes

Reviewer #2: Yes

3. Has the statistical analysis been performed appropriately and rigorously? 

Reviewer #1: Yes

Reviewer #2: Yes

4. Have the authors made all data underlying the findings in their manuscript fully available?

Reviewer #1: Yes

Reviewer #2: Yes

5. Is the manuscript presented in an intelligible fashion and written in standard English?

Reviewer #1: Yes

Reviewer #2: Yes

6. Review Comments to the Author

Reviewer #1: (No Response)

Reviewer #2: Well performed analysis of key metrics to consider in BPH therapy. Further comments on the applicability of this research to other parts of the world would be good to see.

7. PLOS authors have the option to publish the peer review history of their article (what does this mean?). If published, this will include your full peer review and any attached files.

Reviewer #1: No

Reviewer #2: No

---

## [Author Response · Author response to Decision Letter 1]

15 Feb 2022

Dear Dr Rosier,

Many thanks again for your consideration of this manuscript. I have prepared a response to reviewers document that include my answers to all questions and comments. Please let me know if you have additional questions.

Best regards,

Sirikan Rojanasarot

---

## [Editor Report · Decision Letter 2]

3 Mar 2022

PONE-D-21-30773R2A comprehensive analysis of clinical, quality of life, and cost-effectiveness outcomes of key treatment options for benign prostatic hyperplasiaPLOS ONE

Dear Dr. Rojanasarot,

Thank you for submitting your manuscript to PLOS ONE. After careful consideration, we feel that it has merit but does not fully meet PLOS ONE’s publication criteria as it currently stands. Therefore, we invite you to submit a revised version of the manuscript that addresses the points raised during the review process.

ACADEMIC EDITOR: I am very sorry but you still have not corrected the dosage of tamsilosin. Please connect with your co-authors!

We look forward to receiving your revised manuscript.

Kind regards,

Peter F.W.M. Rosier, M.D. PhD

Academic Editor

PLOS ONE
---

## [Author Response · Author response to Decision Letter 2]

11 Mar 2022

Dear Dr. Rosier:

Many thanks again for your suggestions and expertise. I have corrected the dosage of tamsulosin from 40 mg to 400 μg once daily. The co-authors and I cross-checked the dose and confirmed that 400 μg is the dose used in the US. 

We are really grateful for the opportunity to submit our research to PLOS ONE and thank you for your consideration.

Sincerely,

Siri

---

## [Editor Report · Decision Letter 3]

29 Mar 2022

A comprehensive analysis of clinical, quality of life, and cost-effectiveness outcomes of key treatment options for benign prostatic hyperplasia

PONE-D-21-30773R3

Dear Dr. Rojanasarot,

We’re pleased to inform you that your manuscript has been judged scientifically suitable for publication and will be formally accepted for publication once it meets all outstanding technical requirements.

Kind regards,

Peter F.W.M. Rosier, M.D. PhD

Academic Editor

PLOS ONE
---

## [Editor Report · Acceptance letter]

8 Apr 2022

PONE-D-21-30773R3 

A comprehensive analysis of clinical, quality of life, and cost-effectiveness outcomes of key treatment options for benign prostatic hyperplasia 

Dear Dr. Rojanasarot:

I'm pleased to inform you that your manuscript has been deemed suitable for publication in PLOS ONE. Congratulations! Your manuscript is now with our production department. 

Kind regards, 

on behalf of

Dr. Peter F.W.M. Rosier 

Academic Editor

PLOS ONE